# Outrunning LLM Cutoffs: A Live Kernel Crash Resolution Benchmark for All

**Chenxi Huang** [* 1]   **Alex Mathai** [* 1]   **Feiyang Yu** [2]   **Aleksandr Nogikh** [3]   **Petros Maniatis** [4]   **Franjo Ivančić** [5]
**Eugene Wu** [1]   **Kostis Kaffes** [1]   **Junfeng Yang** [1]   **Baishakhi Ray** [1]

## Abstract

Repairing system crashes discovered by kernel fuzzers like Syzkaller is a critical yet underexplored challenge in software engineering. While recent works have introduced Large Language Model (LLM) based agents for Linux kernel crash-resolution, their evaluation benchmarks are usually static and thus, do not capture the evolving nature of the Linux kernel, and suffer from potential data contamination due to LLM knowledge cutoffs. To address the above problem, we present (i) LIVE-KBENCH, an evaluation framework for *self-evolving* benchmarks that continuously scrapes and evaluates agents on freshly discovered kernel bugs, and (ii) KENV, an agent-agnostic standardized crash-resolution environment for kernel compilation, execution, and feedback. This design decouples agent workflows from heavy-weight execution, enabling fair and scalable comparison across diverse agent frameworks under identical conditions.

To this end, we curate an inaugural dataset of $534$ Linux kernel bugs and empirically demonstrate a significant performance gap, with agents achieving up to 25% higher equivalent patch rate on bugs fixed *before* the LLM knowledge cutoff. Using KENV, we benchmark three state-of-the-art agents, showing that they resolve $74\%$ of crashes on the first attempt (plausible patches); however only $\sim 20\%$ of generated patches closely match developer fixes. Additionally, exposing crash resolution feedback improves crash resolution rate by $29\%$. LIVE-KBENCH provides the community

with an evaluation infrastructure for *self-evolving* benchmarks that is both *time and attribute sensitive*; complete with a public dashboard to track agent progress on Linux kernel bugs.

## 1. Introduction

In recent years, software-engineering agents have made rapid progress in user-space development, where automated bug repair is increasingly achievable in modern programming environments (Anthropic, 2025; Google, 2025; Cursor, 2025; GitHub, 2025; Amazon, 2025). This progress, however, does not readily extend to system-level failures. When the Linux kernel—the foundation of much of modern computing infrastructure—crashes, debugging and repair are substantially more challenging due to limited observability, strict execution constraints, the sheer size of the codebase, and high correctness and safety requirements.

At the same time, kernel fuzzers have become highly effective at uncovering such failures. Tools such as Syzkaller (syz, 2025b) use coverage-guided fuzzing to explore diverse kernel execution paths and have reported over $6K$ kernel crashes, many of which expose high direct or indirect risk (Zou et al., 2022) and are assigned CVE identifiers (Bursey et al., 2024). However, the automated and large-scale nature of fuzzing introduces fundamental obstacles for repair and evaluation. Fuzzer-discovered bugs typically come with machine-generated reports and a single proof-of-concept input, leaving vulnerabilities underspecified and allowing many candidate patches that suppress the crash while breaking functionality (Mathai et al., 2025). These challenges are further compounded by the scale of the Linux kernel—over 20M lines of code—which complicates code localization within LLM context limits, and by the high cost of evaluation, as compiling and testing a single kernel patch can take at least 30 minutes.

**Limitations of Existing Work.** Recent efforts have taken initial steps toward kernel crash resolution. Mathai et al. (2024) introduced KBENCHSYZ, a small-scale benchmark for kernel crash repair, and subsequent work proposed LLM-

---

[*]Equal contribution   [1]Department of Computer Science, Columbia University, New York City, NY, USA [2]Department of Computer Science and Engineering, New York University, New York City, NY, USA [3]Google Inc., München, Germany [4]Google DeepMind, Mountain View, CA, USA [5]Google Inc., Princeton, NJ, USA. Correspondence to: Chenxi Huang <chenxi@cs.columbia.edu>, Alex Mathai <alexmathai@cs.columbia.edu>.

*Proceedings of the $43^{rd}$ International Conference on Machine Learning*, Seoul, South Korea. PMLR 306, 2026. Copyright 2026 by the author(s).

---

LIVE-KBENCH artifacts are available at github.com/ARiSE-Lab/live-kbench.

based agents for fixing kernel crashes (Mathai et al., 2025; Singh et al., 2025). In parallel, rapid advances have been made in general-purpose automated program repair agents in user-space settings (Wang et al., 2024; Yang et al., 2024; Trae Research Team et al., 2025). Despite their close conceptual relationship, these lines of work have largely evolved in isolation: modern repair agents have not been systematically evaluated on kernel crash-resolution tasks, and existing studies rely on older, static datasets that are vulnerable to data contamination and do not reflect the constantly evolving Linux kernel. As a result, prior work provides limited insight into the effectiveness of state-of-the-art agents for real-world kernel crash resolution. To this end, we identify two major limitations of existing work.

(i) *Lack of a kernel-scale, agent-agnostic computer-use environment.* Existing benchmarks such as SWE-Bench and KBENCHSYZ (Jimenez et al., 2023; Mathai et al., 2024) standardize scoring but leave the execution environment unspecified, forcing agent developers to construct their own computer-use environments. Thus, each agent defines its own benchmark-specific logic to set up the environment, which involves preparing the correct codebase and artifacts, supporting local testing, and enforcing safety constraints. While this tightly coupled design is workable for user-space, interpreted languages with low execution overhead, it does not scale to Linux kernel crash resolution. Preparing the kernel codebase is costly, testing requires privileged virtualization, and compiling and executing modified kernels is slow and resource-intensive (Mathai et al., 2024). As a result, evaluation cannot be performed directly within an agent's local workflow and must rely on external execution infrastructure. We therefore argue for a *principled decoupling* of agent workflows from heavy-weight computer-use environments: an agent-agnostic environment layer that automates setup and offloads kernel compilation and testing to a scalable execution backend. This separation enables fair comparison across agents by allowing workflows and tool usage to vary, while environment preparation, execution, and feedback are handled uniformly.

(ii) *Distribution shift and data contamination.* Although recent work has proposed kernel-specific approaches (Mathai et al., 2024; 2025; Singh et al., 2025), evaluations rely primarily on older, static datasets (e.g., KBENCHSYZ), with fixes dating back to 2018. The Linux kernel evolves rapidly through refactoring, subsystem changes, and shifting execution contexts, leading to persistent distribution shift. At the same time, older fixes increasingly appear in LLM training data, which risks data contamination. As a result, static benchmarks may overestimate generalization, underscoring the need for time-aware evaluation.

**Our Approach.** To address these challenges, we develop an agent-agnostic crash-resolution environment and a live

evaluation framework for Linux kernel crash resolution.

(I) **KENV** standardizes computer-use environment preparation and execution for crash resolution by initializing the Linux codebase at the correct commit, enforcing execution guardrails, and exposing a single `run_kernel` interface that submits compilation and execution jobs to KGYM-SUITE, a scalable kernel execution platform (Mathai et al., 2025). Thus, by decoupling agent workflows from heavy-weight execution, KENV enables scalable, reproducible, and fair comparisons across agentic frameworks.

(ii) **LIVE-KBENCH** builds on this execution layer by continuously sourcing fresh kernel bugs from public dashboards and mailing lists, filtering for reproducibility, running registered agents via KENV, and evaluating resulting patches along multiple dimensions—including crash resolution, localization accuracy, and similarity to human fixes—before reporting results on a public leaderboard.

**Results.** Using our systems, we curate LIVE-KBENCH-2512: a fresh dataset with the latest Linux bugs. We then use LIVE-KBENCH to measure the difference in performance on bugs *before* and *after* LLM knowledge cutoffs. In our experiments, we show that agents exhibit up to $25\%$ better equivalent patch rate on *older* data (before the cutoff) compared to *fresh* data (after the cutoff). Additionally, we benchmark three open-source agents by providing each agent access to KENV and then running comprehensive evaluations. Our results confirm that, at a first glance, current agents resolve $74\%$ of kernel crashes in the first try (plausible patches), with only $\sim 20\%$ of the patches being exactly equivalent to the kernel developer's patch. We also show that the crash resolution rate (CRR) improves by $25\%$ when agents can access crash resolution feedback using the `run_kernel` tool.

**Contributions.** In summary, we present LIVE-KBENCH—a live benchmark for Linux kernel crash-resolution. LIVE-KBENCH supports reproducible *time-sensitive* evaluation on an evolving system software, combining continuous bug curation with a standardized crash-resolution environment. Our contributions are threefold:

- **A standardized environment for kernel crash resolution evaluation.** We introduce KENV, an agent-agnostic computer-use environment to instantiate reproducible executable setups for Linux kernel crash instances. By standardizing environment preparation, execution, and feedback, KENV enables fair evaluation across agents under identical conditions.
- **A live, executable benchmark of Linux kernel crashes.** We present LIVE-KBENCH, a continuously updated benchmark that curates reproducible Linux kernel crashes suitable for learning-based evaluation. The initial release, LIVE-KBENCH-2512, contains $534$ instances, with the

benchmark designed to evolve as new Linux kernel bugs are discovered.

- **Time-aware and attribute-based evaluation.** LIVE-KBENCH supports evaluation across time and bug attributes, with a public dashboard that enables analysis by fix date, bug type, agent scaffold, and LLM backends. This design enables studies of distribution shift, data contamination, and cross-agent performance that are not possible with static benchmarks.

**Conflict of Interest Disclosure.** The author AN, PM and FI are employed by Google Inc., which leads the development of Gemini models, which were among the ones evaluated in this paper.

## 2. Related Work

**Coding benchmarks.** Since the emergence of LLMs, their application to software engineering tasks has driven the development of numerous evaluation benchmarks. Early benchmarks such as HumanEval and MBPP focused on function-level code synthesis (Chen et al., 2021; Austin et al., 2021), while SWE-bench extended evaluation to repository-level bug resolution in real-world user-space codebases (Jimenez et al., 2023). More recently, Mathai et al. (2024) introduced system-level crash resolution benchmarks for large operating system codebase (Linux kernel). However, most existing benchmarks are static and manually curated, making them susceptible to data contamination and distribution shift. Recent efforts have begun to address this limitation through automatic curation of fresh evaluation data (Jain et al., 2024; Zhang et al., 2025). However, those attempts focus on issue-solving tasks in user-space code repositories, unlike the low-level system crashes in LIVE-KBENCH. Thus, building on this line of work, we introduce LIVE-KBENCH, a *self-evolving* benchmark for Linux kernel crash resolution that continuously ingests newly discovered bugs from Syzbot (syz, 2025a) and supports time-aware and attribute-aware evaluation beyond LLM knowledge cutoffs.

**Agents.** Primitive agents that pack all necessary information into a single prompt have been shown to achieve abysmal performance on benchmarks consisting of repository-level tasks (Mathai et al., 2024; Jimenez et al., 2023). For example, Mathai et al. demonstrated that even when prompting LLMs with the exact oracle files, the crash resolution rate remained very low on the kBenchSyz dataset. Recently, SWE-agent (Yang et al., 2024) sets the new paradigm of autonomous SE agents by equipping LLMs with computer-use tools, and its bash-only scaffold variant (mini-SWE-agent) is currently used to evaluate various LLMs on the SWE-bench dataset (Jimenez et al., 2023; 2025; Yang et al., 2024). Subsequently, there were many follow-up agentic frameworks, including OpenHands (Wang et al., 2024), TRAE (Trae Research Team et al., 2025), Live-SWE-agent (Xia

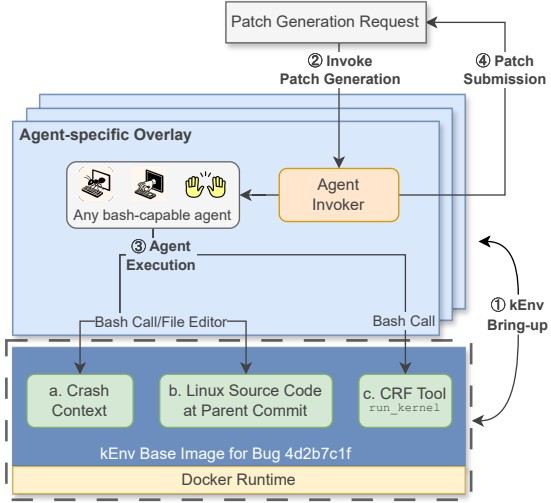

*Figure 1.* Interaction between KENV and LIVE-KBENCH. A KENV instance is brought up and merged with an agentic-specific overlay (①). A patch generation request from LIVE-KBENCH invokes the agent (②), and it runs within the KENV instance (③). Finally, the patch is submitted to LIVE-KBENCH ④.

et al., 2025), CodeResearcher (Singh et al., 2025) and Crash-Fixer (Mathai et al., 2025). In this work, we benchmark the representative coding agents in §5.3 and compare their performance on Linux kernel crash-resolution tasks.

**KGYM.** Upon requests, KGYM (Mathai et al., 2024) schedules subroutines of the kernel building and testing onto a cluster of KGYM workers to scale up the throughput of verification requests. The successor of KGYM, KGYM-SUITE (Mathai et al., 2025), improved scalability and performance to support workflow-based agents such as Crash-Fixer (Mathai et al., 2025). However, this infrastructure was tightly coupled to a specific agent design. To address this limitation, we develop KENV, a truly agent-agnostic environment that connects to KGYMSUITE to provide standardized kernel compilation and execution support. Using KENV, we evaluate three automated program repair agents in §5.3, demonstrating its extensibility across agents.

## 3. KENV: A Standardized Environment for Linux Kernel Crash Resolution

In this section, we present KENV, an agent-agnostic environment that supports any `bash`-capable agent for Linux kernel crash resolution. Rather than tailoring the environment setup to individual agent scaffolds, KENV abstracts this layer to provide a standardized kernel crash-resolution interface, enabling seamless support for diverse agents. We describe what makes KENV agent-agnostic (§3.1) and its support for exposing crash resolution feedback (CRF) (§3.2).

### 3.1. Design

Figure 1 details the KENV system. Upon receiving a patch generation request, KENV brings up the base instance environment and installs the relevant agent as shown in ①.

*Base Image*: In particular, KENV builds a base Docker image for the kernel bug by following these steps: (a) it synthesizes the crash context with the crash report, crash reproducer, and instructions on using the CRF tool (§3.2); (b) it clones the Linux repository and checks out to the right commit ID; and (c) it installs the CRF tool. Once built, this base image is stored for future use (dark blue box in Figure 1).

*Agent Overlay*: This build step is specific to a target agent scaffold. KENV builds the final Docker image by "overlaying" the agent-specific Dockerfile (light blue box in Figure 1) over the base image. The agent-specific Dockerfile describes the agent's installation steps and invocation logic. This final image can now be used to invoke the agent on the kernel crash resolution task (described below).

*Agent Invoker*: A patch generation request from LIVE-KBENCH begins the agent invocation process (②). The agent invoker executes (③) the agent with the crash context and waits for the agent to finish. Inside KENV, the agent can reason about the crash context, explore the Linux source code with tool invocations, and receive crash-resolution feedback using the CRF tool. Once the agent finishes editing the local Linux codebase, the agent invoker will collect all outputs—including the final patch as a `git diff` and metadata (dollar cost, time cost, and agent trajectory); finally returning them to the patch generation request (④).

It is important to note that this process does not change the agentic scaffold. Hence, KENV is agent-agnostic by design. By abstracting out the base image, any agent overlay can access the standardized crash-resolution environment. We use KENV to support mini-SWE-agent, SWE-agent, and OpenHands on LIVE-KBENCH.

### 3.2. Crash Resolution Feedback (CRF)

Agents routinely use regression tests to validate their local edits (FAIR CodeGen team et al., 2025). CrashFixer uses an iterative process of making edits and collecting crash-resolution feedback to substantially improve performance over one-shot prompting (Mathai et al., 2025). Motivated by this, we design the KENV environment to provide crash resolution feedback (CRF) by invoking a scalable kernel experimentation platform through a `run_kernel` bash script (also referred to as "CRF tool" in Figure 1). More specifically, we place the CRF tool and add its description to the agent's task prompt. This interface is *minimal* and *uniform* across agents — involving a simple string manipulation. Internally, a `run_kernel` invocation performs two steps: (a) it generates the agent's edits in *git diff* format, and (b)

composes a crash reproduction job that is offloaded to a remote kernel execution platform. The platform returns crash resolution feedback with the following possible results: (1) a "crash resolved" message, (2) a "crash reproduced" message along with the new crashing stack trace, or (3) a "compilation error" message indicating the current `diff` was not compilable. This mechanism enables CRF for *all* bash-capable agents. To study the impact of CRF on crash resolution rate, we perform ablations in §5.6.

## 4. LIVE-KBENCH: Enabling Self-evolving Kernel Crash Resolution Benchmarks

Prior work on automated kernel crash resolution (Mathai et al., 2024; 2025; Singh et al., 2025) evaluates agents on older benchmarks such as KBENCHSYZ, which fail to capture ongoing kernel evolution — incurring potential concept drift, and are at the risk of data contamination. Motivated by these limitations, we introduce LIVE-KBENCH, a *self-evolving* benchmark that continuously curates fresh kernel bugs and supports multi-faceted agent evaluation. We next describe the design of LIVE-KBENCH (§4.1), its evaluation metrics (§4.2), and the empirical studies it enables (§4.3).

### 4.1. Live Benchmarking

**Data source.** LIVE-KBENCH sources its data from Syzbot (syz, 2025a), an automated system that invokes Syzkaller (syz, 2025b) — an open-source coverage-guided fuzzer that discovers new Linux kernel vulnerabilities. Syzbot (syz, 2025a) reports these kernel vulnerabilities (read "crashes") to the Linux kernel mailing list (LKML) (Nogikh, 2023). This process is continuous, with Syzbot initiating the fuzzing process on the latest Linux repository, and sending reports for newly found bugs each day on the LKML to kernel developers worldwide.

**Data curation.** A weekly-scheduled crawler collects newly discovered Syzbot bugs, and this incoming set of bugs triggers live benchmarking. In the beginning, LIVE-KBENCH populates the metadata for the incoming bug set (❶ in Figure 2). For each bug, LIVE-KBENCH populates metadata such as `git` commit information (e.g., the patch date, patch content, etc), kernel configuration, reproducer, and crash report. We then store this bug metadata in a database. Subsequently, LIVE-KBENCH invokes KGYMSUITE with this metadata to generate SuiteCache (an intermediate compiled state of each kernel). SuiteCache enables incremental compilation of Linux kernels, significantly speeding up kernel compilation time (Mathai et al., 2025). LIVE-KBENCH then runs bug reproduction to filter out unreliable crashes. Due to the nondeterministic nature of kernel bugs, we allow for five reproduction attempts (❷). The reproducible bugs are then finally selected for patch generation.

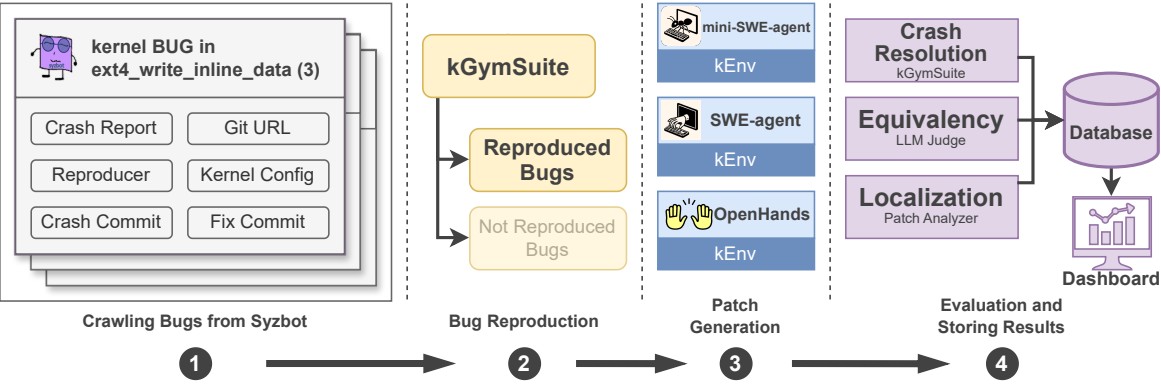

*Figure 2.* Live Benchmarking. LIVE-KBENCH first curates kernel bugs from Syzbot (❶), filters out bugs that are reliably triggered (❷), executes agents on the bugs using KENV (❸), and computes and stores metrics that are finally displayed to a dashboard (❹).

We bootstrap LIVE-KBENCH with bugs reported from April 2024 to December 2025 as an inaugural dataset of 534 bugs — LIVE-KBENCH-2512. We compare LIVE-KBENCH-2512 with previous work and show detailed characteristics in the Appendix (§A).

**Agentic patch generation in KENV.** Once bug curation is complete, LIVE-KBENCH invokes KENV to build the base Docker image for each bug, which is then overlayed with each agentic framework. LIVE-KBENCH then runs each agent and collects the final output — the agent patch, dollar cost, time spent, and tool trajectory (❸). LIVE-KBENCH then proceeds to execute a multi-faceted evaluation of the agent-generated patches (details in §4.2).

### 4.2. Evaluation Metrics

When agents submit their patches through KENV (❹), LIVE-KBENCH evaluates the patches with three metrics.

**(1) Crash resolution.** This metric measures if the patch successfully resolves the crash. LIVE-KBENCH leverages KGYMSUITE for this task. Due to the nondeterministic nature of most kernel crashes, we configure KGYMSUITE to run 25 crash-reproduction runs on the patched kernel to ensure that the fix resolves the crash.

For bugs without the associated developer patch (i.e., open bugs), our evaluation stops here. For patched bugs, we additionally evaluate the agent patch for root-cause localization and patch equivalence (explained below).

**(2) Root cause localization.** We develop a program analysis tool—a patch analyzer that accurately extracts modified files and functions from a patch. To measure the performance of root cause localization, we run the patch analyzer on the agent patches as well as the ground-truth fixes, and calculate intersection-over-union score (**IoU**) at file and function granularities. While previous work on bug localization adopts mean reciprocal rank (MRR) and Recall@k (Mathai et al.,

2024; Zhou et al., 2025), autonomous agents do not rank files or functions when performing patch generation, and no implicit preference can be extracted from the agent's trajectory; hence we resort to **IoU**.

**(3) Equivalence.** Manual studies in previous work (Mathai et al., 2025) that analyze the equivalence between an agent patch and the ground-truth patch are hard to scale. However, equivalence is crucial to measure data contamination as it reflects the LLM's tendency to generate a patch almost identical to the developer's fix. To perform large-scale patch equivalence evaluation in LIVE-KBENCH, we deploy an LLM judge to generate equivalence verdicts. We use the ground-truth fix commit (and commit message) from the kernel developer and run the LLM judge to check for equivalence between the agent patch and the ground-truth patch. For each agent patch, the LLM judge produces multiple votes of "equivalent" or "discrepant", and the verdict is determined by majority voting. The criterion that the LLM judge uses for an equivalence study is rigid: accounting for differences in variable names, two patches are considered equivalent only if their structure and logic are exactly the same. We describe the current LLM judge configuration used in LIVE-KBENCH in §5.1 and evaluate its efficacy with ground-truth manual analysis data in §5.7.

**Caveat.** Crash resolution is a necessary (but insufficient) condition for a valid patch, and equivalence is a sufficient (but unnecessary) condition. Neither these individual metrics nor their combination can *guarantee* validity. Instead, the valid patch rate of an agent lies between the crash resolution rate (upper bound) and the equivalent patch rate (lower bound). This nuance is important, as there exist common patterns in crash-resolving yet invalid patches (Mathai et al., 2025) such as (1) amputation (i.e., bypassing faulty code altogether) and (2) significantly modifying existing functionality. We further discuss the challenge of evaluating kernel patches in §8.

### 4.3. Empirical Studies Enabled by LIVE-κBENCH

An intriguing benefit of an automated and live kernel bug benchmarking system is that it enables the community to perform interesting empirical studies on kernel crash resolution. This is enabled by a key distinguishing factor, the *co-existence* of reproducible kernel bugs, their rich crash metadata, and the evaluation metrics of various agents on these bugs on a *single unified* dashboard (❹). Before we explain how this is done, we first list the available attributes that can be used to filter and query relevant bug subsets.

**Views.** LIVE-κBENCH enables multiple data views through its dashboard (shown in Appendix §6), such as:

(1) Kernel bugs filtered by fixed time, subsystem, bug type, cause bisection. Additionally, if the patch is available, then dev patch size, dev patch modified files, lingering time (time between the discovery and fix), fix commit, amongst others.

(2) Agent patch evaluation results filtered by scaffold; test-time scaling (if multiple runs exist); κENV parameters such as crash resolution feedback (on or off), localization assistance (oracle or non-oracle), cost limit (concrete limit or unlimited); trajectory attributes such as dollar cost and time spent; and patch attributes such as crash resolution, patch equivalence, localization IoU, agent patch modified files, agent patch modified functions , amongst others.

**Empirical Questions.** Equipped with this rich dashboard, it is possible to address interesting empirical questions:

- Is there a performance difference for LLM-based agents on data *before* and *after* the knowledge cutoff?
- What are the "toughest bugs" in terms of crash resolutions that are not solved by *any* agent?
- What are the crash resolution or patch equivalence rates in terms of bug type, subsystem, or other attributes?
- How unique is the evolution of Linux kernel bugs, and does it periodically reflect distinct concept drifts?

To get solutions to such questions, we can use our framework. For instance, to perform the first experiment, one can filter fixed bugs by their fixed time into two sets— before and after the cutoff (LIVE-κBENCH ✔), run the agent of choice (κENV ✔), and finally aggregate and compare metrics for both runs (LIVE-κBENCH ✔). To answer the second question, one can curate a set of bugs (LIVE-κBENCH ✔), run a *family* of agents (κENV ✔), and aggregate the crash resolved count; finally filtering for bugs where count is zero (LIVE-κBENCH ✔).

While the list of empirical questions is endless, we focus on the following research questions (given our resource constraints) and report the details in §5:

- What is the performance difference of agents on kernel crash resolution tasks across the LLM knowledge cutoff? (RQ1 in §5.2)
- How does the choice of agentic scaffold impact kernel crash resolution performance? (RQ2 in §5.3)
- How do current LLMs perform on kernel crash resolution, and what is the impact of test-time scaling? (RQ3 in §5.4)
- What is the upper-bound performance on kernel crash resolution tasks with perfect localization? (RQ4 in §5.5)
- Is crash resolution feedback beneficial for autonomous agents? (RQ5 in §5.6)

## 5. Experiments

In this section, we showcase several highlighted empirical studies enabled by LIVE-κBENCH. We elaborate on our general setup (§5.1), and report our experimental details in §5.2, §5.3, §5.4 and §5.5.

### 5.1. LIVE-κBENCH Setup

**Dataset.** We use LIVE-κBENCH-2512 as our primary dataset since it contains relatively fresh data (reported since April 2024, fixed since June 2024) up until December 2025.

**Metrics.** As discussed in §4.2, we report crash resolution rate (**CRR**) which is the percentage of bugs with resolved crashes (see more details in Appendix Section C); equivalent patch rate (**EPR**) which is the percentage of bugs with perfectly equivalent *and* crash-resolving patches; and the average intersection over union (**IoU**) ratio of individual patches to measure the localization performance of agents, at both the file and function granularities.

**Agentic scaffolds.** We use the representative autonomous agent systems mini-SWE-agent (Yang et al., 2024), SWE-agent (Yang et al., 2024), and OpenHands (Wang et al., 2024) for evaluation. Due to the efficacy of crash resolution feedback (CRF) shown in §5.6, we enable CRF in κENV for agents to test their patches during runtime.

**LLMs.** We use Gemini 3 Pro, Gemini 3 Flash, Claude Sonnet 4.5 and Claude Opus 4.5 to measure performance with state-of-the-art LLMs in our experiments. The published knowledge cutoff date of Gemini 3 models and Claude Sonnet 4.5 is January 2025.

**LLM Judge.** In the experiments we configure LLM Judge with Gemini 3 Flash to sample 9 votes, and a patch is considered equivalent to its corresponding developer's patch when 5 or more votes support the "equivalent" verdict. We evaluate this LLM judge configuration in §5.7.

### 5.2. RQ1: Performance difference spanning cutoff

**Setup.** In this RQ, we use LIVE-κBENCH to measure the performance of Gemini 3 Pro on LIVE-κBENCH-2512 split by the LLM knowledge cutoff date. We describe below an initial setup of LIVE-κBENCH for Gemini 3 Pro which one

*Table 1.* Performance of mini-SWE-agent and Gemini 3 Pro under limited budget, split at the knowledge cutoff date (January 2025)

| Split | Before Cutoff | After Cutoff |
|---|---|---|
| **Pass@1** | | |
| CRR (%) | 78.44 (↑8.72%) | 72.15 |
| EPR (%) | 15.60 (↑20.28%) | 12.97 |
| **Pass@10** | | |
| CRR (%) | 92.20 (↑3.69%) | 88.92 |
| EPR (%) | 33.03 (↑25.73%) | 26.27 |
| **Mean@10** | | |
| CRR (%) | 77.84 (↑6.85%) | 72.85 |
| EPR (%) | 16.74 (↑22.73%) | 13.64 |

*Table 2.* Performance of mini-SWE-agent and Claude Sonnet 4.5 (temperature = 1, budget = $5.6 budget), split at the knowledge cutoff date (January 2025)

| Split | Before Cutoff | After Cutoff |
|---|---|---|
| **Mean@3** | | |
| CRR (%) | 67.85 (↑9.77%) | 61.81 |
| EPR (%) | 13.30 (↑17.80%) | 11.29 |

can also repeat it in LIVE-KBENCH across many other LLM backends and agentic scaffolds (given enough resources).

We compare the CRR and EPR on two subsets of the LIVE-KBENCH-2512 dataset in Figure 3 (orange and green lines). These subsets are constructed by splitting LIVE-KBENCH-2512 based on Gemini's knowledge cutoff date. The first set is bugs with *fix commits* on or before January 31, 2025, and the second set has bugs fixed after this date. We collect CRR and EPR on each subset and calculate the change in model performance between these subsets. Additionally, we also include Claude Sonnet 4.5 with budget constraint ($5.6) to extend this experiment to a different model family. As mentioned earlier, we chose to use Gemini and Claude Sonnet as the LLM backend and refrained from repeating experiments with other LLMs due to resource limitations.

**Temporal Splits.** We compare the pre- and post-cutoff subsets along two dimensions — subsystem coverage and size of developers' patch. For subsystem coverage, 38 of 46 pre-cutoff subsystems (83%) also appear in the post-cutoff set; the 8 pre-cutoff-only subsystems account for 13 bugs (6%), and the 24 post-cutoff-only subsystems account for 47 bugs (15%). For patch size, both sets show similar proportions of small (1–5 lines: 42% pre, 47% post) and medium (6–10 lines: 18% pre, 16% post) fixes. Based on these features, we observe no notable distributional concerns between the two splits.

**Results.** In Figure 3 and Table 1, on Gemini 3 Pro, we observe a significant performance advantage in both CRR (↑6.85%) and EPR (↑22.73%) on data before the knowl-

edge cutoff. The advantage in terms of CRR shrinks from (↑8.72%) at Pass 1 to (↑3.69%) at Pass 10 as the number of attempts increases, while the EPR advantage grows from (↑20.28%) to (↑25.73%). In Table 2, we find similar patterns for Claude Sonnet 4.5. Recalling the criterion for patch equivalency, two patches are considered equivalent if their structure and logic are the same (barring minor variable naming differences). Therefore, the EPR advantage suggests that more agent patches are **"almost the same"** to their ground-truth counterpart for data *before* the cutoff (rather than after).

### 5.3. RQ2: Efficacy of agentic scaffolds

**Setup.** We run controlled experiments in LIVE-KBENCH with mini-SWE-agent, SWE-agent and OpenHands on Gemini 3 Pro for three runs, and report the evaluation results in Table 4. For SWE-agent, we enable the bash tool, the `str_replace_editor` tool, and the search tool. For OpenHands, we adopt CodeActAgent configuration with the code execution tool enabled and the browsing tool disabled. We run agents with an unlimited budget and unrestricted number of LLM API calls.

**Results.** We report the performance of different agentic scaffolds in Table 4. In terms of CRR, SWE-agent trails behind mini-SWE-agent and OpenHands. However, its EPR is the highest among the three. In general however, all three agents seem to have relatively similar performance — this is not surprising as they share similar key agentic design choices. The consistently low EPR (∼15%) across agents suggests that achieving substantial further improvements may require fundamentally different approaches.

### 5.4. RQ3: Efficacy of LLM backends

*Table 3.* Performance of mini-SWE-agent with LLMs on LIVE-KBENCH-2512 under $5.6 budget constraint. Gemini results are averaged over three runs (temperature = 1).

| Model | Gemini 3 Flash | Gemini 3 Pro | Claude Opus 4.5 |
|---|---|---|---|
| **CRR (%)** | 48.63 | 70.22 | 74.16 |
| **EPR (%)** | 8.18 | 14.67 | 19.85 |
| **File Loc. IoU (%)** | 51.66 | 64.23 | 69.15 |
| **Func. Loc. IoU (%)** | 33.61 | 45.52 | 53.53 |

In Table 3, we report CRR, EPR and IoU (file/function level) for state-of-the-art LLMs paired with the mini-SWE-agent scaffold on kernel crash resolution. For Gemini, we sample three times by setting the temperature to 1 and reporting the average across runs. For Claude Opus 4.5, we run it once with the temperature set to 0 per common practice in previous software engineering benchmarks (Jimenez et al., 2023; Zhang et al., 2025). Claude Opus 4.5 leads in kernel crash resolution tasks with higher scores across all met-

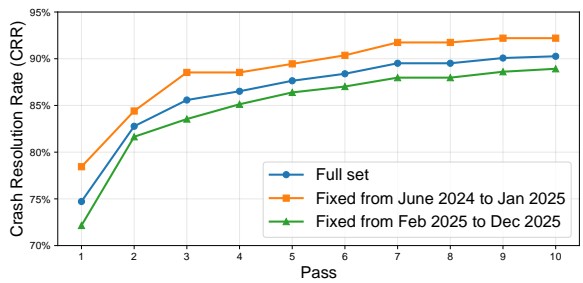 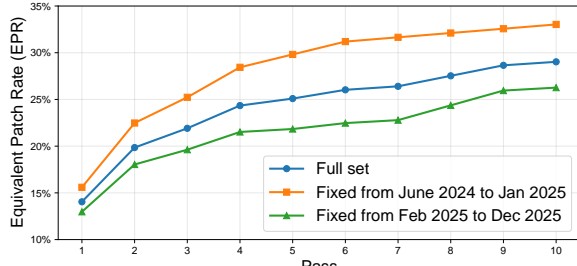

*Figure 3.* Pass@k performance of mini-SWE-agent with Gemini 3 Pro on LIVE-κBENCH-2512 and its subsets before and after the knowledge cutoff date (Jan 2025) with unlimited budget.

*Table 4.* Mean@3 Performance of representative agentic frameworks with Gemini 3 Pro under unlimited budget

| Agent | mini-SWE-agent | SWE-agent | OpenHands |
|---|---|---|---|
| **CRR** | 75.03% | 72.47% | 72.91% |
| **EPR** | 15.29% | 15.86% | 14.73% |
| **File IoU** | 64.35% | 65.89% | 64.95% |
| **Func. IoU** | 45.66% | 47.12% | 46.33% |

rics. This assumes a maximum budget of $5.6 for each bug. We set this limit by analyzing the 95th percentile of the cost for each bug (averaged across 10 runs) when running mini-SWE-agent with Gemini-3-Pro in an unlimited budget setting ($5.55). The full cost distribution is in Appendix §4.

**Test-time scaling.** For each bug in LIVE-κBENCH-2512, we perform ten independent attempts at patch generation using mini-SWE-agent with Gemini 3 Pro. Figure 3 shows the test-time scaling performance on the whole set of LIVE-κBENCH-2512 (blue line). As one would expect, with test-time scaling, both CRR and EPR grow steadily increasing up to 90% (↑21.42%) and 30% (↑109.45%) respectively.

### 5.5. RQ4: Performance with perfect localization

**Setup.** To measure the upper-bound crash resolution performance with perfect localization, we provide the ground-truth modified files to agents (i.e. oracle mode) and report CRR and EPR under such setting. We run two agents on LIVE-κBENCH-2512 in oracle mode—mini-SWE-agent and CrashFixer [1] (an agent that runs only in oracle mode, Mathai et al. (2025)).

**Results.** In Table 5 we report the upper-bound performance by providing the set of files modified by the kernel developers. When enabling oracle mode for mini-SWE-agent, there is a drop of 6.32% in CRR but 12.30% increase in EPR. This shows that perfect localization assistance improves the precision of mini-SWE-agent, as the portion of equivalent patches within crash resolving patches grows. We also observe lower CRR with a higher EPR, which means less crash

---

[1]CrashFixer is not open-sourced, but the authors agreed to run it on LIVE-κBENCH-2512 and provide patches (not trajectories)

suppression given the perfect file localization.

*Table 5.* Mean@3 performance of mini-SWE-agent and Crash-Fixer in oracle mode and mini-SWE-agent in non-oracle mode on Gemini 3 Pro under unlimited budget

| Agent | CRR (%) | EPR (%) |
|---|---|---|
| CrashFixer | 68.79 | 16.54 |
| mini-SWE-agent | 69.35 | 17.17 |
| mini-SWE-agent (non-oracle) | 75.03 | 15.29 |

### 5.6. RQ5: Performance impact of CRF

**Setup.** Prior work (Mathai et al., 2025) demonstrated that iterative editing guided by crash-resolution feedback (CRF) can improve performance in a workflow-based agent. We revisit this idea to evaluate the impact of CRF for a general autonomous agent by comparing mini-SWE-agent in κENV with CRF enabled and disabled, and analyzing the associated system cost.

**Performance impact.** As shown in Table 7, when CRF is enabled, the CRR increases dramatically (↑29.12%). However, we also point out that despite achieving a higher CRR with CRF enabled, the EPR and localization IoU metrics stay unchanged. This suggests that there is still work to do to produce valid patches (equivalent to those of a developer).

To study whether the agents actually called this tool during these runs, we depict the average number of run_kernel invocations in Table 6. As can be seen, the average number of invocations is 1.7, indicating that mini-SWE-agent is motivated to refine its edit when receiving negative feedback on crash resolution.

**Cost of CRF.** CRF requires a significant amount of CPU cycles and a virtualization environment to compile and test a modified kernel. In Table 6, we show the impact of CRF on trajectory time usage. Compared to the average total time spent during LLM inference (4.68 minutes), the average aggregated time during the run_kernel invocations is 7× (33.12 minutes). Hence, grounding patches in execution feedback demands high CPU usage. According to GCP

*Table 6.* Statistics of LLM and `run_kernel` invocations from ten runs of mini-SWE-agent trajectories with Gemini 3 Pro under unlimited budget on LIVE-kBENCH-2512

| Per Invocation | Avg. | Stddev |
|---|---|---|
| LLM Latency (second) | 8.34 | 14.71 |
| `run_kernel` Latency (minute) | | |
| ↪ When Given Compilable patch | 19.98 | 5.05 |
| ↪ When Given Uncompilable patch | 1.97 | 0.67 |
| ↪ Overall | 18.71 | 6.71 |
| **Per Trajectory** | **Avg.** | **Stddev** |
| # LLM Invocation | 33.67 | 19.54 |
| # `run_kernel` Invocation | 1.77 | 1.730 |
| Aggregated LLM Latency (minute) | 4.68 | 3.44 |
| Aggregated `run_kernel` Latency (minute) | 33.12 | 24.75 |

*Table 7.* Performance of mini-SWE-agent and Gemini 3 Pro under unlimited budget vs. CRF

| Configuration | Without CRF | With CRF |
|---|---|---|
| **Pass@3** | | |
| CRR (%) | 71.54 | 85.58 (↑19.63%) |
| EPR (%) | 21.54 | 21.91 (↑1.72%) |
| **Mean@3** | | |
| CRR (%) | 58.11 | 75.03 (↑29.12%) |
| EPR (%) | 15.04 | 15.29 (↑1.66%) |
| Loc. IoU (%) | 64.91 | 64.35 (↓0.86%) |
| Func. Loc. IoU (%) | 45.68 | 45.66 (↓0.04%) |

Compute Engine pricing, we calculated an estimated cost for each `run_kernel` job as $0.28 (see more details in Appendix §B).

**Crash suppression.** Crash resolution feedback is inherently symptom-oriented: it rewards any patch that prevents the reproducer from crashing, regardless of semantic correctness. This may incentivize shallow suppression strategies such as inserting early returns or bypassing assertions. Empirically, as shown in Table 7, CRF improves CRR more than EPR, confirming that a portion of additionally resolved crashes are attributable to suppression rather than genuine repair.

### 5.7. LLM Judge

In §5 we conduct experiments using an LLM judge with a specific configuration. Mathai et al. (2025) includes a small-scale manual comparison of their agent-generated patches against developer-written patches for 79 bugs in kBENCH-SYZ. To measure the efficacy of our LLM judge, we contacted the authors and requested these manual comparisons. We then ran our LLM judge on the agent-generated patches and obtained an alignment score between our LLM judge and their manual observations. We note that the LLM judge has an impressive accuracy of ∼ 90% and a F1 score of 83%, and we show more details in Appendix (Section D).

## 6. Discussions

**Crash-resolving but non-equivalent patches.** We identify three common failure modes: *(i) Amputation / early return*: the agent patch directly steers the control flow to avoid triggering the assertion or sanitizer, effectively suppressing the symptom without addressing the root cause. *(ii) Silent error*: rather than returning an appropriate error code, the agent patch modifies a variable to a valid value to suppress the crash silently. *(iii) Non-coherent hotfix*: the agent patch addresses the crash but does not maintain proper semantics of other states or corresponding error-handling logic. These patterns highlight a recurring tendency of agents to produce shallow, symptom-level fixes that satisfy the crash reproducer without achieving semantic correctness.

**Non-crash-resolving patches.** Of the 52 bugs that remain unsolved after 10 CRF-enabled mini-SWE-agent runs with Gemini 3 Pro, we identify two dominant patterns: *(i) Partial fix*: the agent patch only fixes the component partially, and the reproducer triggers a different crash in the same component. *(ii) Latent bug surfacing*: the agent patch resolves the original crash, but the reproducer triggers a separate, pre-existing bug elsewhere in the system. These cases illustrate an inherent challenge of crash-based evaluation in a large codebase: resolving one fault may expose another, making CRR a necessary but not sufficient signal for patch correctness.

## 7. Conclusion

In this paper, we present LIVE-kBENCH, a *self-evolving* benchmark for Linux kernel bugs. We also introduce KENV, an agent-agnostic standardized environment for kernel crash resolution. We curate an inaugural dataset (LIVE-kBENCH-2512) and perform time-aware and attribute-based empirical studies. Our experiments show that agents perform 25% better in terms of EPR on bugs fixed before the LLM knowledge cutoff (than after), state-of-the-art LLM with agentic scaffold resolves 74% of crashes with only ∼ 20% EPR, and that crash resolution feedback improves CRR by 29%. We hope LIVE-kBENCH to enable the research community advancing research in Linux kernel crash resolution and its evaluation.

## 8. Future Work

There exist several directions to advance Linux kernel crash resolution such as (a) defining a new metric (other than CRR and EPR) to accurately measure patch efficacy; (b) extracting execution information to provide a dynamic view of the crash behavior (in addition to the current static information), and (c) lightweight solutions to optimize feedback time when using the CRF tool.

## Acknowledgments

This research received support from Google, NSF CCF-2313055, as well as DAPLab corporate support in the form of funding and/or compute from Amazon, IntellectAI, Infosys, Tidalwave, Veris, Shopify, Microsoft, Thinking Machines, Dandy, Perplexity, and Daytona. The views and conclusions presented here are those of the authors and should not be interpreted as representing the official positions of the funding organizations.

## Impact Statement

This paper presents work aimed at advancing machine learning methods for automated software engineering, particularly in the evaluation of coding agents on complex, real-world system software. By providing a standardized, reproducible benchmark and execution environment, our work seeks to improve the rigor and transparency with which coding agents are evaluated.

As with many advances in automated programming, our work could have both positive and negative societal implications. On the positive side, improved evaluation and understanding of coding agents may contribute to more reliable software development tools, faster debugging of critical infrastructure, and reduced human effort in maintaining complex systems. At the same time, more capable automated code-modification systems could potentially be misused if applied without appropriate safeguards.

We do not introduce new learning algorithms or deployable agents, but rather focus on evaluation infrastructure and benchmarking. As such, we believe the risks associated with this work are limited and comparable to those of prior software engineering benchmarks. Nonetheless, we encourage responsible use of automated coding systems and emphasize that our benchmark is designed for research and evaluation purposes only.

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

# Appendix

## A. Characteristics of LIVE-KBENCH-2512

The detailed fundamentals of LIVE-KBENCH-2512 are shown in Table 9, and we compare our inaugural dataset — LIVE-KBENCH-2512 with previous works in Table 8. We show the distribution of LIVE-KBENCH-2512 in Figure 5 in terms of bug type and fixed month spanning from June 2024 to December 2025.

*Table 8.* Comparison of LIVE-KBENCH-2512 and previous work (Mathai et al., 2024; Jimenez et al., 2023; Zhang et al., 2025)

| Benchmark | Total | Before Cutoff | After Cutoff | Average Repo. Size (LoC) |
|---|---|---|---|---|
| **LIVE-KBENCH-2512** | 534 | 218 | 316 | > 20,000,000 |
| KBENCHSYZ | 279 | 279 | 0 | > 20,000,000 |
| KBENCHC | 504 | 504 | 0 | > 20,000,000 |
| SWE-bench-Live (full) | 1888 | 1087 | 801 | ∼ 85,000 |

Knowledge cutoff (Gemini 3 Pro): January 2025
SWE-bench-Live statistics queried on Jan 19, 2026

*Table 9.* Dataset card of LIVE-KBENCH-2512

| LIVE-KBENCH-2512 | |
|---|---|
| # Bug Instances | 534 |
| # Kernel Subsystems | 70 |
| # Bug Types | 10 |
| Avg. Fixed bugs Per Month | 28.10 |
| Avg. Size of Gold Patch | |
| ↪ # LoC | 16.57 |
| ↪ # Files | 1.48 |
| Median Days from Reported to Fixed | 11 |

## B. CRF Cost Estimation

We host KGYMSUITE on Google Cloud Platform (GCP), allocating c3 vCPUs for kernel builder (kBuilder), n2 vCPUs as well as e2 vCPUs for kernel testing (kVMManager). We collect the statistics of all `run_kernel` jobs on KGYM-SUITE in Table 10 and estimate the cost of each CRF call as $0.28, using GCP pricing on Jan 29, 2026 (gcp, 2025).

*Table 10.* Cost breakdown of KGYMSUITE deployed on GCP

| Component | Avg. vCPU Time | vCPUs | GCP Pricing ($/vCPU) | Total |
|---|---|---|---|---|
| **kBuilder** | 545.05 seconds | 4× c3 | 4.43 | $0.28 |
| **kVMManager** | 576.16 seconds | 4× n2 | 1.55 | |

## C. Crash Resolution Evaluation Parameters

Due to the indeterministic nature of kernel crash reproduction, we run crash reproduction multiple times on KGYM-SUITE. For crash resolution evaluation, following previous work (Mathai et al., 2024; 2025), we run 25 times of crash reproduction to ensure low probability of persistent false negative. However, in CRF, to lower the latency in the agentic runtime, we run 5 times of crash reproduction for `run_kernel` calls.

To quantify the effect of the crash reproduction budget, we measure CRR at Pass@$k$ for $k = 5, 10, 15, 20, 25$ on patches generated by mini-SWE-agent with Gemini 3 Pro (5,340 patches across 10 runs of 534 bugs). Table 11 reports the marginal number of newly reproduced crashes and the corresponding CRR delta per 5-run increment. Both quantities decline monotonically, indicating that CRR is converging: each additional batch of 5 runs surfaces fewer previously undetected nondeterministic reproductions. Extrapolating the trend, doubling the budget to Pass@50 would shift CRR by an estimated 2–3 percentage points at most, well within the noise floor of inter-agent performance differences in our main results. We note that $k=25$ already represents the maximum our computational resources permit, as each run requires executing the modified kernel 25 times across all 5,340 cases; nevertheless, the convergence trend suggests that additional runs beyond 25 would yield diminishing returns.

*Table 11.* Convergence of crash reproduction with increasing Pass@$k$ budget (mini-SWE-agent, Gemini 3 Pro, 5,340 patches).

| Increment | New Crashes | Δ CRR (%) |
|---|---|---|
| $k$: 5 → 10 | 90 | 1.70 |
| $k$: 10 → 15 | 81 | 1.52 |
| $k$: 15 → 20 | 63 | 1.19 |
| $k$: 20 → 25 | 51 | 0.96 |

## D. LLM Judge Performance

In Table 12, we benchmark different models on LLM judge alignment with the manual analysis data from CrashFixer. As shown, Gemini 3 Flash is the best among others, and therefore we use Gemini 3 Flash as the LLM judge model in our experiments. In Table 13, we show more details of the performance of the LLM judge (with Gemini 3 Flash as LLM backend) used in §5 on manual analysis data from CrashFixer.

*Table 12.* Performance of LLMs on LLM Judge alignment with CrashFixer patches

| Model | Accuracy (%) | F1 (%) |
|---|---|---|
| Gemini 3 Pro Preview | 86.08 | 77.55 |
| Claude Opus 4.5 | 87.34 | 78.26 |
| Gemini 3 Flash Preview | **89.87** | **83.33** |
| Gemini 2.5 Pro | 86.08 | 77.55 |

*Table 13.* Performance of LLM Judge (Gemini 3 Flash) on Crash-Fixer patches

| True Positive | 20 | True Negative | 51 |
|---|---|---|---|
| False Positive | 3 | False Negative | 5 |
| Accuracy | 89.87% | Precision | 86.96% |
| Recall | 80.00% | F1 | 83.33% |

# E. Other Tables and Infographics

We plot the distribution of the average cost of ten mini-SWE-agent runs with Gemini 3 Pro in Figure 4.

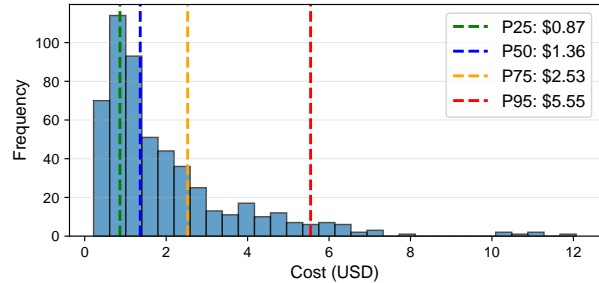

*Figure 4.* Distribution of average cost for each bug of ten mini-SWE-agent runs with Gemini 3 Pro with unlimited budget

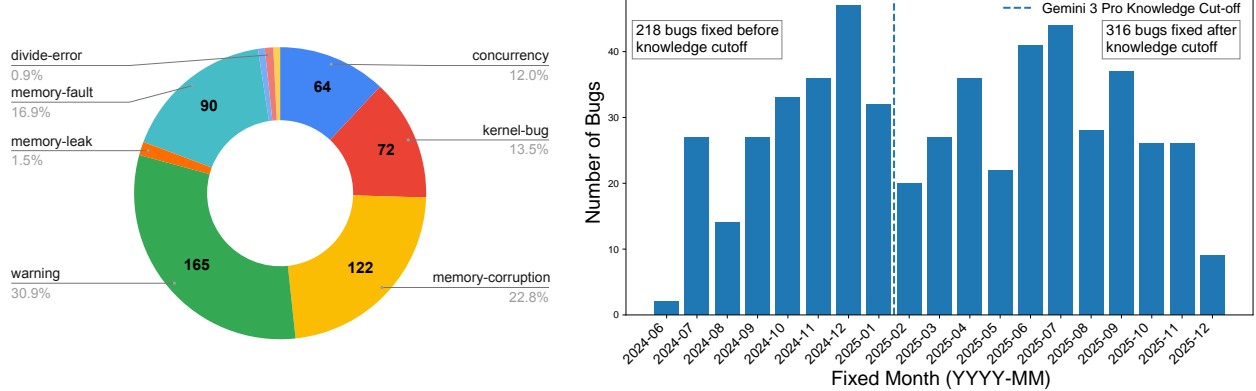

*Figure 5.* Distributions of LIVE-KBENCH-2512 (534 bugs)

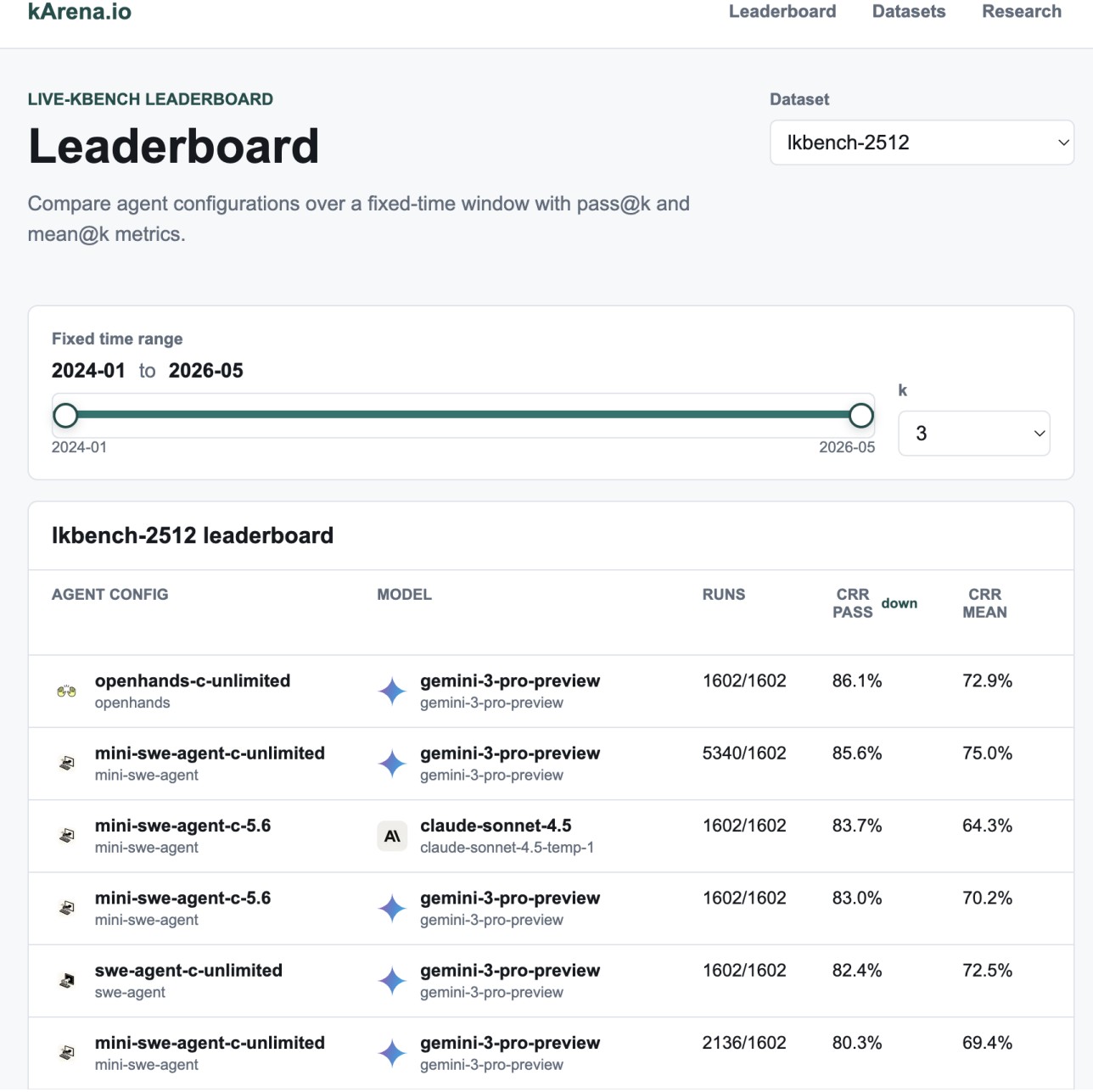

*Figure 6.* Screenshot of LIVE-KBENCH Dashboard

