# OpenReview forum: "Outrunning LLM Cutoffs: A Live Kernel Crash Resolution Benchmark for All"
_ICML.cc/2026/Conference — ICML 2026 regular_

### Official Review · Reviewer_eJn8 · 2026-03-12

**Soundness:** 2
**Presentation:** 3
**Significance:** 3
**Originality:** 2
**Overall Recommendation:** 3
**Confidence:** 4

**Summary:**

This paper proposes a new evaluation framework named Live-kBENCH for evaluating LLM agents’ ability to fix kernel vulnerabilities. Live-kBENCH has three features: 1) a standard evaluation environment named kEnv that provides a dedicated Docker image for each kernel vulnerability, 2) a dataset containing 534 kernel vulnerabilities, which can be continuously expanded as new vulnerabilities are reported by syzbot, and 3) a dashboard that allows users to explore and analyze benchmarking results. The paper also conducts time-aware and attribute-based evaluations to analyze agent performance under different settings, such as knowledge cutoff, agentic scaffolds, and model backends.

**Compliance With Llm Reviewing Policy:**

Affirmed.

**Final Justification:**

The final response still cannot address the following concerns.

1. Crash resolve rate is not a good metric (referring to the Arxiv paper does not resolve this concern). Also, the other metric (LLM judged semantic equivalence with the original patch) has low accuracy and F1 for a ground-truth metric (accuracy ∼90%, F1 83%).

2. For the kernel remote build and execution platform, the technique is not clearly explained. The paper just listed some technical challenges, and then just said it's building on kGymSuite with bug fixes, new agent-facing APIs, and hardening for sustained traffic.

**Key Questions For Authors:**

1. The current evaluation metrics may not provide a fully comprehensive assessment. Crash resolution may be somewhat loose, as passing it may only indicate that the patch mitigates a single PoC rather than fixing the vulnerability. Meanwhile, equivalence relies on an LLM judge (accuracy ∼90%, F1 83%), which may not be ideal as a ground-truth oracle, and it may also be too strict since valid fixes could differ from the ground-truth patch. Could the authors consider more fine-grained metrics, such as functionality tests to ensure the patch does not break existing behavior, or fuzzing for some time?

2. Could the authors clarify the technical novelty of the system design? kEnv appears to prepare an independent base image for each kernel vulnerability, and each agent provides its own Dockerfile that builds on top of this image. This seems to be a commonly used engineering approach (e.g., a similar design is in Terminal-Bench). The dashboard also appears more like a usability feature.

3. Section 3.2 mentions a remote kernel execution platform, but its design and necessity are not clearly described. Could the authors provide more details about its architecture and why it is necessary?

4. Section 5.2 compares EPR before and after the knowledge cutoff. However, since EPR is determined by an LLM judge, it may also be influenced by the judge model’s knowledge cutoff. How is this effect separated in the evaluation?

**Limitations:**

yes

**Strengths And Weaknesses:**

Strengths
1. The paper provides a new benchmark for an important and relatively underexplored problem: evaluating LLM agents’ ability to repair Linux kernel vulnerabilities.

Weaknesses
1. The current evaluation metrics may not fully capture patch quality. Crash resolution may be too permissive, while equivalence relies on an LLM judge and may be overly strict.
2. The technical novelty of the system design is unclear.

---

> ### Author Rebuttal · Authors · 2026-03-31
>
> We thank the reviewer for the valuable feedback.
>
> 1. Before answering this question, we expand on the current state of affairs for testing in the Linux kernel.
>
>    1. Existing test suites have very low coverage. As per [Wang LPC’24](https://lpc.events/event/18/contributions/1793/attachments/1624/3447/LPC'24 Linux Testing.pdf), kUnitTests (unit tests for Linux) cover only ~2% of Linux code. With both “Linux Testing Project” (LTP) and the “Kselftest” testsuites, coverage is still ~35%.
>    2. Targeted test creation and targeted fuzzing for Linux is still an open and challenging problem. Current approaches require careful manual curation of 1000's of system call specifications [[Tan CCS’23](https://dl.acm.org/doi/10.1145/3576915.3623146)] and are not currently well-automated.
>
>    As the aim of this question is to generate or leverage test cases that exercise the code paths modified by the patch, this is basically a “targeted” testing/fuzzing problem. As mentioned above, this is a very difficult task considering the limited observability and the limited methods we can leverage on the Linux kernel. We would also mention that we had previously set up a clean functional pipeline of the LTP testsuite. We confirmed that, though we ran ~3500 tests, there was almost *zero* execution intersection with the patch-modified code. As such, creating “relevant” tests (aside from the PoC) is an area of future work.
>
> 2. We agree that the containerization pattern (per-task base images with agent-defined Dockerfiles) is a standard engineering approach and is not, by itself, the source of novelty (e.g., similar patterns exist in Terminal-Bench). The dashboard is also a usability feature rather than a core contribution.
>    The **technical novelty of kEnv lies in how execution is exposed to the agent for kernel-level tasks**, which differ fundamentally from typical userspace software benchmarks. In this setting, validation is **compute-intensive and stateful**: each iteration requires compiling the kernel, booting it, and running a reproducer to observe crash behavior and stack traces. To support this, kEnv introduces a **domain-specific execution oracle (`run_kernel`) as a first-class agent tool**, decoupled from the agent’s local environment. Instead of arbitrary terminal interaction, the agent queries a **standardized interface** that returns one of three types of outcomes: crash resolved, crash reproduced (with a stack trace), or compilation error. This enables **structured reasoning over execution outcomes** and abstracts away the complexity of kernel build/boot pipelines.
>    This design is qualitatively different from Terminal-Bench (unstructured terminal access). The importance of this structured, high-cost oracle is validated by the **29% CRR improvement from enabling CRF** (Table 6), showing that it is critical for kernel crash resolution.
>
> 3. The evaluation of a kernel patch is vital for patch evaluation and for crash resolution feedback (CRF). However, evaluating kernels “at scale” requires building a distributed system to run compute-intensive processes in “parallel”– compiling modified kernel versions and running lengthy bug reproducers (>10 minutes); far harder than running a single line of `python -m pytest`. Unlike the Terminal-Bench benchmark, where agents are able to run tests (or interact with the execution environment) in a container, the same approach is not scalable for kernel benchmarks. There is a need to separate the “CPU-light” work (e.g. the agent searching for code or making edits to files) from the “CPU-heavy” work (i.e. building kernels and running kernel tests). With this design, we were able to launch hundreds of lightweight Docker instances on a single machine, run agents simultaneously on these Docker instances, and, when each agent demands CRF to check its current patch, it issues a `run_kernel` job that is dispatched to the remote system. This allows a “smaller” remote execution system (~30 nodes)–consisting of expensive and high-performance computers to be well utilized by hundreds of agent instances. Adopting the approach taken by Terminal-Bench, though simpler in system design, is prohibitively expensive and inefficient, as the need for high-performance computers would scale linearly to the number of parallel agents. Additionally, by abstracting away this execution capability in kEnv, we provide easy access to the execution platform for **all** agents.
>
> 4. The judge used in our experiments is an “equivalence” judge instead of a standard uninformed LLM judge. It makes informed decisions (as it has access to a reference ground truth) and judges the equivalence between the ground-truth patch and the agent patch. The LLM judge is provided with agent patch, developer patch and developer’s Git commit message to make informed verdict on comparison task. Such a “comparison” task is not affected by knowledge cutoffs, as the verdict remains unchanged even if the LLM has seen the ground-truth patch before.

---

> > ### Author Rebuttal · Reviewer_eJn8 · 2026-04-03
> >
> > Thanks for the response. (1) Regarding evaluation metrics, the rebuttal justifies the difficulty of creating fine-grained tests, but this does not address the core concern that the current metrics remain coarse-grained and may not fully capture patch quality. (2) Regarding the remote execution system, the rebuttal describes the motivation for decoupling agent and kernel building/testing environments, but the non-trivial technical challenges of building this system remain unclear. (3) Regarding run_kernel, the rebuttal describes its interface and outputs but still lacks sufficient implementation details.

---

> > > ### Author Response · Authors · 2026-04-07
> > >
> > > We appreciate you pointing out the concerns in our submission and for your constructive discussion. At the outset, before addressing the questions, we would like to assure you that our source code will be open-sourced, so any questions regarding our exact implementation methodology can be readily verified against our software artifacts.
> > >
> > > 1. We appreciate the reviewer's continued engagement on this point. We acknowledge that CRR and EPR represent two different standards of patch quality — CRR is the minimal necessary condition for a good patch, and EPR is the most rigorous sufficient condition. However, we point out that **together** they are more than just two coarse numbers—they provide a meaningful interval that bounds the true valid patch rate, and the relationship between them across experimental conditions reveals insights about agent behavior.
> > >
> > >    As a concrete example, Reviewer ihTj correctly observed that in oracle mode (Table 4), CRR drops by ~6% while EPR increases by ~12% when agents are restricted to editing **only** the files modified by human developers. From this observation, we infer that **unrestricted** **agents** may be circumventing crashing behavior via out-of-scope modifications—a form of crash suppression rather than genuine repair. This is a valuable research insight about how LLM agents occasionally approach kernel crash resolution, and it was derived directly and easily from analyzing the interplay between CRR and EPR across experimental settings on Live-kBench. No single metric, however fine-grained, would have surfaced this — it emerged from having both metrics together under controlled conditions. We believe this demonstrates the value of Live-kBench as a platform for understanding agent behavior.
> > >
> > >    We would also want to clarify that proposing/defending a metric is **not** a contribution in this work — it is an auxiliary outcome of the benchmark. The CRR metric has been used by related work like CodeResearcher and CrashFixer and both papers acknowledge that it is an upper bound to real efficacy. We have additionally added EPR as a lower bound for the same. Besides, we have also made an honest attempt to leverage existing techniques and test suites (as they stand today and as we described earlier) and have confirmed that they are not useful in judging the patches of the vulnerabilities exposed by Syzkaller.
> > >
> > > 2. We thank the reviewer for the clarifying question and take this opportunity to expand on the remote execution platform. As you suspected, there are significant technical challenges underlying the remote execution platform. First, **older kernels** routinely fail with modern toolchains, requiring transparent backporting of fix-up commits. Second, **from-scratch** **kernel builds** take tens of minutes, necessitating aggressive caching and a fast pre-build path for early error detection. Third, **execution** requires provisioning VMs, transferring kernel images, booting, injecting reproducers, and parsing serial console output. Fourth, a single benchmark evaluation may invoke the execution platform hundreds of times across multiple agents and seeds, demanding **distributed scheduling** with proper isolation.
> > >
> > >    We address these by building on kGymSuite (the open-source successor to kGym from CrashFixer), which provides worker decomposition and RabbitMQ-based scheduling. We extended it with bug fixes, new agent-facing APIs (incremental patch application, build-cache introspection, per-attempt resource accounting), and hardening for sustained traffic. The resulting cluster runs on 30 high-performance GCP instances, exposed uniformly through a single `run_kernel` entry point so agent authors can focus on reasoning rather than systems plumbing.
> > >
> > > 3. `run_kernel` tool is implemented as a bash script that culminates with an API call to the remote execution platform. The tool gathers agent’s modifications through git diff, retrieves the corresponding kernel compilation cache for extremely fast compilation, and composes a bug reproduction job request. `run_kernel` later polls the remote execution platform for job progress. Once the remote execution platform processes the request, `run_kernel` will retrieve the job result (one of “not reproduced”, “crashed with a new crash report”, or “compilation error”) and show it as standard output to the agent.

---

### Official Review · Reviewer_ihTj · 2026-03-12

**Soundness:** 3
**Presentation:** 3
**Significance:** 3
**Originality:** 3
**Overall Recommendation:** 4
**Confidence:** 4

**Summary:**

This paper introduces LIVE-KBENCH, a continuously evolving benchmark for evaluating LLM-based agents on Linux kernel crash resolution, and KENV, an agent-agnostic execution environment that decouples heavyweight kernel compilation and testing from agent workflows. The benchmark ingests fresh bugs from Syzbot to mitigate data contamination from LLM training cutoffs. An inaugural dataset of 534 bugs is curated, and several state-of-the-art agent frameworks and LLMs are evaluated. This work is a valuable contribution toward assessing how software engineering agents handle complex, real-world systems.

**Compliance With Llm Reviewing Policy:**

Affirmed.

**Final Justification:**

The authors have addressed partial of my concerns. I have raised my score accordingly.

**Key Questions For Authors:**

1. Can you provide a matched analysis of the temporal split to show whether pre- and post-cutoff bugs differ in size, subsystem coverage, or complexity?
2. What do the non-equivalent, crash-resolving patches look like? A qualitative categorization (e.g., "amputations" vs. "alternative fixes") would be highly informative.
3. Why does oracle localization reduce the CRR? Can you characterize the out-of-scope modifications made by unrestricted agents?
4. Have you validated the LLM judge on any LIVE-KBENCH-2512 instances using manual expert annotation or other strategies?

**Limitations:**

Yes

**Strengths And Weaknesses:**

**Soundness**

1. The knowledge-cutoff analysis attributes the post-cutoff CRR and EPR decline to data contamination. However, this assumes that bug difficulty remains stationary across the temporal split. Without controlling for bug-type distribution, patch complexity, or subsystem coverage, the observed decline could simply reflect an increase in task difficulty. Additionally, the analysis is limited to a single LLM and agent scaffold, which restricts the generalizability of the findings. More rigorous experimentation is needed to confirm the contamination claim.
2. The CRF evaluation appears to conflate "crash suppression" with genuine repair. Under iterative feedback, agents may be incentivized to stop a crash by any means necessary, such as inserting early returns, removing assertions, or commenting out code. Furthermore, running a reproducer 25 times may be insufficient to capture nondeterministic race conditions. The oracle-mode results in Table 4 reinforce this concern: restricting agents to the correct files decreased the CRR by ~6% while increasing the EPR by ~12%. This suggests that unrestricted agents may circumvent the crash test via out-of-scope modifications.
3. The LLM judge uses Gemini 3 Flash to evaluate patches from stronger models. Kernel patch equivalence is extremely subtle; minor changes in spinlocks or memory allocation flags can introduce deadlocks while appearing textually insignificant. The reported 90% accuracy/83% F1 score rests on 79 samples from a different dataset, leaving the judge's reliability on novel, post-cutoff bugs unvalidated. The authors do not justify using a single LLM over a diverse ensemble, nor do they explore non-LLM approaches like AST matching or semantic diffs.

**Presentation**

The paper is clearly written and logically organized around research questions. Architecture diagrams effectively convey the pipeline.

**Significance**
An agent-agnostic benchmark for kernel repair fills a genuine gap in the field. The infrastructure is valuable even if the empirical claims are currently contested. However, its ultimate significance is tied to the reliability of its metrics, which, as current evidence suggests, may not yet accurately measure repair quality.

**Originality**

Adapting live benchmarking to the Linux kernel, given the engineering hurdles of virtualization and nondeterministic reproduction, is a meaningful advancement over existing user-space benchmarking efforts. While the empirical findings may not be particularly surprising, the work provides a valuable infrastructure that is crucial for future research.

---

> ### Author Rebuttal · Authors · 2026-03-31
>
> We thank the reviewer for the feedback and the important questions.
>
> 1. We had measured distribution similarity when performing RQ1 and confirmed that both datasets have very similar characteristics. We list some statistics here:
>
>    1. Subsystem coverage is broadly consistent. 38 of 46 pre-cutoff subsystems (83%) also appear in the post-cutoff set, covering all major areas: networking (net, netfilter, wireless), filesystems (ext4, btrfs, bcachefs, f2fs, etc.), memory management (mm), and core kernel components. The 8 subsystems only in the pre-cutoff period, account for just 13 bugs total (6% of 218). The 24 subsystems in post-cutoff-only subsystems likewise represent a small fraction — 47 of 316 bugs (15%) — spread thinly across niche areas like comedi, block, and kvm.
>    2. Complexity is comparable. Both sets show similar proportions of small (1–5 lines: ~42% pre, ~47% post) and medium (6–10 lines: ~18.3% pre, ~16.1% post) fixes. Neither set is systematically harder or simpler by this measure.
>
>    Overall, the two splits are well-matched: they share the same dominant subsystems, nearly identical patch size distributions, and comparable complexity profiles. The post-cutoff set is larger (316 vs. 218 bugs) but not qualitatively different, supporting the validity of using the temporal split to assess generalization.
>
> 2. Through manual analysis on a subset of non-equivalent, crash-resolving patches, we categorize the failure modes of these patches as following:
>
>    1. Amputation / early return: the agent patch directly steers the control flow to avoid triggering the assertion / sanitizer.
>    2. Silent error: instead of returning an error code, the agent patch directly modifies a variable to a valid value to suppress the crash.
>    3. Non-coherent hotfix: the agent patch deals with the crash but does not maintain the proper semantics of other states or corresponding error handling.
>
> 3. Since the execution path from the reproducer to the crash spans across many functions in the Linux kernel, an agent can modify any function in this execution chain to prevent a crash. From the example patches above, when given no oracle file set, we show that agent patches can resolve a crash by performing early-return and amputating normal functionality in other files.
>
> 4. We (the authors) randomly selected 100 patches from the agents' predictions, conducted a manual analysis to study the equivalence between the developer and agent patches, and finally compared this to the judge’s verdict. We found Gemini 3 Flash to have an accuracy of 88% and an F1 score of 70%, roughly in line with results.
>
> Additionally, to respond to the reviewer’s concern in Soundness (1), we agree that concretely investigating the effect of data contamination in LLMs requires substantially more experimentation—ideally generating multiple batches of agent patches across two sets of bugs matched in distribution of multiple dimensions (e.g., difficulty, subsystem, patch size, etc.). This presents three key challenges: (1) scaling up experiments across different agentic scaffolds and LLM configurations, (2) accumulating a sufficient number of kernel bugs to enable sampling of comparably distributed subsets, and (3) securing the LLM and CPU computing resources needed for inference and patch evaluation, including multiple independent runs to establish statistical confidence.
>
> In this paper, we address these challenges as follows: kEnv targets challenge (1), Live-kBench and its ongoing accumulation target challenge (2), and within our limited computing budget, RQ1—which examines the performance gap of Gemini 3 Pro across its knowledge cutoff date using kGym and Live-kBench—serves as an initial investigation into the broader problem of data contamination. Moreover, to initially approach challenge (3), in RQ1, we investigate the performance gap for Gemini 3 Pro on temporal splits with similar subsystem coverage and distribution of developer patch size. To be clear, RQ1 does not claim to infer the effects of data contamination; rather, it documents an observable performance gap for Gemini 3 Pro specifically spanning the cutoff and demonstrates that Live-kBench enables the community to gather further empirical evidence through easy-to-use, fast, and scalable evaluations.

---

> > ### Author Rebuttal · Reviewer_ihTj · 2026-04-04
> >
> > Thanks the authors for their feedback. I have raised the score of soundness.

---

> > > ### Author Response · Authors · 2026-04-07
> > >
> > > We want to thank the reviewer for the positive feedback! To resolve any further questions, we have added the following information from our previous experiments to address the remaining concerns:
> > >
> > > 1. **Crash suppression in CRF.** As agents can run `pytest` to check if basic functionalities of a patched software work, we introduce CRF in the picture to bridge the gap between kernel-level software engineering task and test-time patch testing through one-click of `run_kernel`. We totally agree that the unrestricted agents may circumvent the crash test via out-of-scope modifications, and the evaluation result of CRR decreasing and EPR increasing in oracle mode is precisely an example that researchers can observe through enabling or disabling CRF our Live-kBench benchmarking system.
> > > 2. **Pass@25 for bug reproduction.** We appreciate the reviewer's concern about nondeterministic crash reproduction. We follow previous work [2] when evaluating crash resolution rate by running a reproduction test for 25 times. To quantify the effect, we measured CRR at pass@k for k = 5, 10, 15, 20, 25 for patches generated by mini-swe-agent on Gemini 3 Pro. Out of 5340 patches (10 runs of 534 bugs), the marginal number of newly reproduced crashes per 5-run increment is 90 (k=5->10), 81 (k=10->15), 63 (k=15->20), and 51 (k=20->25), showing a clear and consistent decline. Correspondingly, the CRR delta shrinks from 1.70% (k=5->10) to 1.52% (k=10->15) to 1.19% (k=15->20) to 0.96% (k=20->25). This monotonically diminishing marginal rate indicates that the CRR is converging: each additional batch of 5 runs surfaces fewer previously-undetected flaky reproductions. Extrapolating the trend, even doubling the budget to pass@50 would shift the CRR by an estimated ~2–3 pp at most — well within the noise floor of inter-agent performance differences reported in our main results. We note that k=25 already represents the maximum our computational resources permit, as each reproducer run requires executing the modified kernel for 25 times, and the full evaluation spans 5340 cases. Nevertheless, the convergence trend strongly suggests that additional runs beyond 25 would yield diminishing returns, and we consider this a reasonable cost–confidence trade-off that captures the vast majority of nondeterministic failures.
> > > 3. **Gemini 3 Flash as the model for LLM judge.** We adopt the design of majority voting and high temperature from one of the prevalent LLM judge designs [1]. We then evaluated the LLM judge with different LLM backends on the 79 human-annotated data we received from the CrashFixer authors. As shown below, we chose Gemini 3 Flash as it performed the best. Note that the 79 human-annotated dataset is a subset of kBenchSyz, which is also a curated set of Syzkaller bugs, though from 2018 to 2023. Moreover, in the previous response (Answer #4 in the rebuttal), we did manual analysis on 100 patches randomly sampled, and we found decent agreement (88% accuracy, 70% F1) between LLM Judge (Gemini 3 Flash) and human annotators on Live-kBench dataset.
> > >
> > > | Backend                | Accuracy | F1    |
> > > | ---------------------- | -------- | ----- |
> > > | Gemini 3 Pro Preview   | 86.08    | 77.55 |
> > > | Claude Opus 4.5        | 87.34    | 78.26 |
> > > | Gemini 3 Flash Preview | 89.87    | 83.33 |
> > > | Gemini 2.5 Pro         | 86.08    | 77.55 |
> > >
> > > Source:
> > > - [1] [A Survey on LLM-as-a-Judge](https://arxiv.org/abs/2411.15594)
> > > - [2] [CrashFixer](https://arxiv.org/pdf/2504.20412)

---

### Official Review · Reviewer_GUcW · 2026-03-13

**Soundness:** 3
**Presentation:** 3
**Significance:** 3
**Originality:** 3
**Overall Recommendation:** 4
**Confidence:** 3

**Summary:**

This paper studies the problem of automatically repairing Linux kernel crashes using LLM-based agents. Kernel repair is substantially harder than standard bug fixing because the Linux kernel has a very large codebase, limited observability, and expensive compilation and testing cycles. Existing evaluations rely on static benchmarks, which are vulnerable to data contamination: models may simply reproduce fixes already seen during training rather than demonstrate real debugging and repair ability.

To address this, the paper introduces LIVE-KBENCH, a self-evolving benchmark that continuously collects fresh kernel bugs from automated testing systems such as syzbot/syzkaller, allowing agents to be evaluated on genuinely new crashes. It also proposes KENV, an agent-agnostic environment that standardizes compilation, execution, and testing of kernel patches, enabling fair comparison across agents under identical conditions. The paper benchmarks multiple agents on a curated set of kernel bugs and argues that current static benchmarks overestimate agent capability due to contamination effects.

**Compliance With Llm Reviewing Policy:**

Affirmed.

**Key Questions For Authors:**

How sensitive are the main conclusions to model choice? In particular, would the contamination gap in RQ1 remain similar if at least one additional model family were included?

Can the authors provide more detailed failure analysis for unresolved crashes, especially those that remain unsolved even with CRF?

How should practitioners think about the compute-performance tradeoff of CRF in real deployment scenarios? Is the reported gain worth the additional runtime and CPU cost?

How robust is the LLM-judge-based equivalence metric to semantically correct but structurally different patches?

The paper does not analyze why specific patches succeed or fail The practical importance of reported increase in EQR from 20% to 25% for real world agents not fully discussed

RQ2 limitations Notes low EPR, does not explore why agents fail to match developer patches.

RQ3 For Claude Opus, only a single run with temp 0 was performed, which can bias results Gemini is run at a different temperature = 1, Opus at temperature = 0. Differences in stochasticity may affect the comparison, but not discussed.

RQ5 failure analysis is missing. The paper does not explore which crashes remain unsolved despite CRF. The cost of CRF is mentioned in terms of CPU time but the trade off is not fully analyzed. For instance, a 29% gain in crash resolution is worth a 7x increase in runtime?

**Limitations:**

Yes

**Strengths And Weaknesses:**

Strengths:
The paper tackles a difficult and underexplored problem: automatic Linux kernel crash repair is both practically important and technically challenging.

LIVE-KBENCH is a strong contribution. Unlike static benchmarks, it continuously collects fresh bugs, making evaluation more realistic and reducing contamination from model pretraining data.

The paper provides convincing evidence that data contamination is a serious issue. The reported performance gap between older potentially seen bugs and fresh unseen bugs suggests that current agent evaluations may overestimate true generalization.

KENV is an important engineering contribution. By standardizing kernel compilation, execution, and feedback in an agent-agnostic way, it makes comparison across different agents much more reliable and reproducible.

The public dashboard is a useful addition for transparency, reproducibility, and long-term progress tracking by the research community.

Weaknesses
The paper does not sufficiently analyze why patches succeed or fail. Several sections report performance numbers, but there is limited failure analysis explaining which crashes remain unsolved and why.

The discussion of CRF is incomplete. While it improves performance, it also consumes substantially more resources. The paper does not adequately discuss whether the tradeoff is practical for real-world automated repair settings.

The long-term sustainability of LIVE-KBENCH is somewhat uncertain. Since it depends on the continued availability of syzbot/syzkaller data and ongoing maintenance of the crawler, its long-term impact may depend on active upkeep.

The LLM-judge-based equivalence criterion may be too rigid. If two patches are considered equivalent only when their structure and logic closely match, the evaluation may unfairly penalize alternative but correct fixes.

The writing could be clearer. Some sections, especially the descriptions of LIVE-KBENCH and KENV, are dense and harder to follow.

Terms such as data contamination, distribution shift, and Docker runtime could be clarified more clearly for readability.

The paper could provide more intuition for why CRF improves performance, beyond describing the technical steps.

Certain experimental comparisons may not be fully fair. For example, some models are run with different temperature settings, and Claude Opus is evaluated with only a single run at temperature 0, which may affect comparability.

The practical significance of some reported gains is not fully discussed. For instance, improvements in repair metrics are reported, but the real-world meaning of these gains for automated development workflows remains somewhat unclear.

---

> ### Author Rebuttal · Authors · 2026-03-31
>
> We thank the reviewer for the feedback and the important questions.
>
> 1. Model sensitivity. Due to resource constraints, we conducted an additional experiment. We ran mini-swe-agent on Claude Sonnet 4.5 with a maximum budget of $5.6 at temperature 1 for 3 independent runs. We observe similar performance gap (Mean@3, knowledge cutoff: Jan 2025):
>
> | Metric % | Before | After |
> | -------- | ------ | ----- |
> | CRR      | 67.89  | 61.81 |
> | EPR      | 13.30  | 11.29 |
> | File IoU | 52.52  | 54.23 |
> | Func IoU | 37.31  | 38.24 |
>
> 2. Failure analysis for CRF. There are 52 bugs remaining unsolved despite 10 runs with CRF-enabled mini-swe-agent on Gemini 3 Pro. We manually analyze the persisted crashes and identify the following patterns:
>    1. Agent patch only fixes the component *partially* and the reproducer triggers another crash in the same component.
>    2. Agent patch fixes the crash but the reproducer triggers another unrelated bug existing in the system.
> 3. Tradeoff of CRF. CRF mirrors standard kernel development practice: routinely obtain crash resolution feedback, either by compiling and testing locally or by submitting to syzbot via `#syz test`. As shown in Table 5, coding agents spend an average of 33.12 minutes on `run_kernel` across 1.77 invocations—well within what human developers perform when manually solving a bug. We also note that the most painful downside of submitting an “unverified” patch to the Linux kernel is the time wasted by kernel expert reviewing an obviously incorrect patch. This resource (expert review) is the biggest bottleneck in kernel development–and as such, should be minimized at all costs.
> 4. Robustness of LLM judge. Unlike common uses of LLMs as generic quality judges, our LLM judge is explicitly prompted to assess **semantic equivalence between generated and ground-truth patches**, using a rubric focused on **logical and behavioral alignment**. If two code snippets are not structurally identical but use similar logic, they are still considered as equivalent by the judge. Below is an example that the LLM judge marked as equivalent. The human developer creates a temporary variable, performs a conditional check on this variable, prints a `WARN` statement, and finally returns an error code (`-EIO`) on the erroneous path. The agent patch chooses to compare against the function’s return directly, performs an equivalent conditional check, prints a different `WARN` statement, and also returns an error code (though a different one, `-EFSCORRUPTED`) on the erroneous path.
>
> ```
> fs/udf/super.c
> <Developer>
> -	unsigned int table_len;
> +	unsigned int table_len, part_map_count;
> ...
>  	if (ret)
>  		goto out_bh;
> -	ret = udf_sb_alloc_partition_maps(sb, le32_to_cpu(lvd->numPartitionMaps));
> +
> +	part_map_count = le32_to_cpu(lvd->numPartitionMaps);
> +	if (part_map_count > table_len / sizeof(struct genericPartitionMap1)) {
> +		udf_err(sb, "error loading logical volume descriptor: "
> +			"Too many partition maps (%u > %u)\n", part_map_count,
> +			table_len / (unsigned)sizeof(struct genericPartitionMap1));
> +		ret = -EIO;
> +		goto out_bh;
> +	}
> +	ret = udf_sb_alloc_partition_maps(sb, part_map_count);
>  	if (ret)
>  		goto out_bh;
>
> <Agent>
>  	if (ret)
>  		goto out_bh;
> +
> +	if (le32_to_cpu(lvd->numPartitionMaps) > table_len /
> +						 sizeof(struct genericPartitionMap)) {
> +		udf_err(sb, "error loading logical volume descriptor: "
> +			"Partition table too short for %u partition maps\n",
> +			le32_to_cpu(lvd->numPartitionMaps));
> +		ret = -EFSCORRUPTED;
> +		goto out_bh;
> +	}
> +
>  	ret = udf_sb_alloc_partition_maps(sb, le32_to_cpu(lvd->numPartitionMaps));
>  	if (ret)
>  		goto out_bh;
> ```
>
> 5. Patch failure pattern. We conclude the failure patterns for patches which do not resolve the crash in the answer to question 2. Besides, through manual analysis, we categorize some failure modes of crash-resolving but non-equivalent patches as following:
>    1. Amputation: directly steering the control flow to avoid triggering the assertion
>    2. Silent error: instead of returning an error code, directly assigning a variable with a value that suppresses the crash
>    3. Non-coherent hotfix: dealing with the crash without maintaining the proper semantics of other states or corresponding error handling
> 6. EPR increases across temporal splits. Based on our analysis of common agent failure patterns, **data contamination is a likely primary driver** of the ~20% increase, though **additional confounding factors may also contribute**.
> 7. Failing patterns in EPR. Please see our answer to question 5.
> 8. Model temperature. It is common practice to set Claude model at temperature 0 for software engineering benchmarks (e.g. SWE-bench, SWE-bench goes live, etc.), while Google strongly recommends to use temperature 1 for Gemini 3 Pro. Nevertheless, we run Claude Sonnet 4.5 with the same setting as Gemini 3 Pro. Please see the results in our answer to question 1.
> 9. Tradeoff of CRF. Please see our answer to question 3.

---

> > ### Author Rebuttal · Reviewer_GUcW · 2026-04-01
> >
> > Thank you for explaining all details. Satisfactory updates. Will keep the same score. Thank you.

---

> > > ### Author Response · Authors · 2026-04-01
> > >
> > > Thank you for taking the time to review our rebuttal and for confirming that your concerns have been adequately addressed. We truly appreciate your thoughtful engagement with our work.
> > >
> > > We noticed that you selected "Fully resolved" and mentioned that our updates were satisfactory, which is very encouraging. We were wondering if there might be any additional improvements — whether to the paper itself, the experiments, or the presentation — that could further strengthen the submission in your view and potentially warrant a score adjustment *at the current rebuttal phase*.
> > >
> > > We are happy to incorporate any further suggestions you may have. Thank you again for your valuable feedback.

---

### Decision · Program_Chairs · 2026-04-30

**Decision:**

Accept (regular)

**Comment:**

This paper introduces Live-kBench and kEnv, an evolving benchmark and execution environment for evaluating LLMs on Linux kernel crash resolution. By continuously sourcing fresh bugs, it effectively addresses the critical issue of data contamination in static benchmarks.

The reviewers provided leaning-positive scores: two Weak Accepts (4) and one Weak Reject (3). All reviewers agreed that automated kernel repair is a highly important, underexplored problem and praised the infrastructure's community value. The primary debates centered on the evaluation metrics (Crash Resolution Rate and Equivalent Patch Rate), with concerns that they might not fully capture true patch quality and could reward mere "crash suppression." Reviewer eJn8 also questioned the technical depth of the execution environment.

During the rebuttal, the authors provided strong clarifications. They included new model ablations, detailed failure analyses, and convincingly explained how the interplay of CRR and EPR actually helps expose agent behaviors like crash suppression. They also effectively clarified the heavy-lifting engineering behind the remote execution platform.

While one reviewer maintained reservations about metric granularity, the majority were satisfied. Given the pressing need for dynamic, contamination-resistant benchmarks in the LLM era, the practical utility of this infrastructure outweighs its metric imperfections. I recommend a Weak Accept.